

# Analysis of the rhizosphere bacterial diversity of *Angelica dahurica* var. *formosana* from different experimental sites and varieties (strains)

Meiyan Jiang, Fei Yao, Yunshu Yang, Yang Zhou, Kai Hou, Yinyin Chen, Dongju Feng and Wei Wu

Sichuan Agricultural University, Chengdu, China

## ABSTRACT

**Background:** Rhizosphere bacteria play important roles in plant growth and secondary metabolite accumulation. Moreover, only with favorable production areas and desirable germplasm can high-yield and high-quality medicinal materials be produced. However, whether origin and germplasm indirectly affect the yield and quality of *Angelica dahurica* var. *formosana* through rhizosphere bacterial effects are not known.

**Methods:** In this study, a high-throughput sequencing strategy was used to explore the relationship between the rhizosphere bacterial community and the cultivation of *A. dahurica* var. *formosana* from different production areas and germplasm for the first time.

**Results:** (1) *Proteobacteria* was the dominant bacterial phylum in the rhizosphere soil of *A. dahurica* var. *formosana*, and these bacteria were stable and conserved to a certain extent. (2) High abundance of *Proteobacteria* was an important rhizospheric indicator of high yield, and high abundance of *Firmicutes* was an important indicator of high quality. *Proteobacteria* and *Firmicutes* might have an important relationship with the yield and quality of *A. dahurica* var. *formosana*, respectively. (3) PCoA cluster analysis demonstrated that both production area and germplasm affected the bacterial community structure in the rhizosphere of *A. dahurica* var. *formosana* to a certain extent, and production area had the greatest effect. In addition to available potassium, the rhizosphere soil nutrient levels of different production areas strongly affected the bacterial diversity and community. These findings provide a theoretical basis for the exploitation and utilization of rhizosphere microbial resources of *A. dahurica* var. *formosana* and offer a novel approach for increasing the yield and quality of this crop.

## INTRODUCTION

The rhizosphere is a specific microecosystem for soil–root–microorganism interactions, which is formed by different plant, soil and environments. Moreover, the rhizosphere microbial community is considered the second genome of plants (*Berendsen, Pieterse &*

Corresponding author
Wei Wu, ewuwei@sicau.edu.cn

*Bakker, 2012*). Plant roots can customize their rhizosphere microbial community through secretions, which in turn can affect plant growth through mineral nutrient absorption, antagonism, nutrition, spatial site competition, soil improvement and plant-induced systemic resistance (*Dutta & Podile, 2010*). Moreover, a variety of biotic and abiotic factors affect the assembly of the rhizosphere microbial community. The specific factors that play decisive roles vary by soil type and plant species. The stronger the environmental screening effect of a factor is, the greater its relative contribution to microbial community assembly (*de Ridder-Duine et al., 2005*). Therefore, understanding the key factors affecting the composition and assembly of beneficial bacterial communities in the rhizosphere may be a sustainable and effective way to actively manage soil microbial communities, improve soil conditions, reduce the application of pesticides and fertilizers, and promote plant growth (*Glick, 2010*).

*Angelica dahurica* var. *formosana* (known as Baizhi in Chinese), which is produced in Sichuan Province, is a famous medicinal plant, particularly in the Suining region of Sichuan Province. It is not only used medicinally in curing colds, headaches, nasal congestion, runny nose, and toothache but also possesses high commercial value for its wide applications in the production of food, health care products, spices, skin care and beauty products and other items. Modern pharmacological studies have found that coumarins, as the main active components of *A. dahurica* var. *formosana*, have anti-inflammatory (*García-Argáez et al., 2000*), antibacterial (*Suleimenov, 2009*), vasodilator (*Wang et al., 2016*), anticancer (*Kim, Kim & Ryu, 2007*), antiviral and antioxidant effects (*Bai et al., 2016*), and the levels of two coumarins, imperatorin and isoimperatorin, are regarded as the most important indices to determine the quality of *A. dahurica* var. *formosana*. There is a long history of the cultivation of *A. dahurica* var. *formosana* in Suining of Sichuan Province with high yield and excellent quality. However, some challenges associated with *A. dahurica* var. *formosana* production, such as early bolting, decreased production areas, and excessive use of pesticides and fertilizers, have led to declines in total yield in recent years.

In addition to suitable cultivation and management methods, good germplasm and suitable environments are crucial factors for effective medicinal plant cultivation (*Shen et al., 2017*; *Li et al., 2020*; *Beschoren da Costa et al., 2022*). Moreover, plants with better germplasm recruit more suitable microbes to promote their own growth (*Bouffaud et al., 2012*; *Wen et al., 2020*). To solve problems associated with production declines and ensure quality, we first explored the beneficial bacterial community in the rhizosphere of *A. dahurica* var. *formosana* from different germplasms and environments to identify organisms linked to high yield and quality. Realizing the potential of the rhizosphere bacterial resources of *A. dahurica* var. *formosana* is considered a good approach to improve the soil environment, reduce the application of fertilizers to a certain extent, and expand the planting region.

Therefore, in this study, the bacterial communities in the rhizosphere soils of *A. dahurica* var. *formosana* from different experimental sites or varieties (strains) were characterized under the same culture management method by using a high-throughput 16S rRNA gene amplicon sequencing strategy. The five experimental sites were Shunjiang

(XA), Shunhe (XB), Sangshulin (XC) and Yongyi (XD) from Suining, the main production areas of *A. dahurica* var. *formosana*, and Chongzhou from Chengdu. The six varieties (strains) were BZA001 (XD), BZA002 (XE), BZA003 (XF), BZA004 (XG), BZB002 (XH) and BZB003 (XI), which were stable varieties (strains) with different growth periods, yields and active ingredient contents. This study may serve as a cornerstone for further research on the biological factors that can improve the quality of *A. dahurica* var. *formosana*. The objectives of the present study were (i) to characterize the rhizosphere bacterial community in samples of *A. dahurica* var. *formosana* from different sites or varieties (strains); (ii) to predict which bacterial communities are closely related to the yield and quality of *A. dahurica* var. *formosana*; and (iii) to determine the influence of experimental sites and varieties (strains) on the bacterial community structure and confirm the importance of good production areas and good germplasm resources in cultivation from the perspective of rhizosphere bacteria.

## MATERIALS AND METHODS

### Experimental design

One of the main *A. dahurica* var. *formosana* production region is Suining of Sichuan Province, China. In previous experiments, our research group bred several varieties (strains) of *A. dahurica* var. *formosana*, which were named BZA001, BZA002, BZA003, BZA004, BZB002 and BZB003. The variety of BZA001 was certified by the Sichuan Crop Variety Certification Committee. The other strains are stable strains of different types obtained by farmers in Sichuan Province after more than 10 years of selective breeding.

To explore the effects of different planting sites for the same varieties (strains) on the rhizosphere bacteria of *A. dahurica* var. *formosana*, BZA001 was used as the test variety along with four different planting sites in Suining and one in Chongzhou as the control with the same cultivation and management methods. The samples from the five different experimental sites were named XA (Shunjiang, Suining), XB (Shunhe, Suining), XC (Sangshulin, Suining), XD (Yongyi, Suining), and XJ (Chongzhou, Chengdu) (Fig. 1). Moreover, to explore the rhizosphere bacteria of different varieties (strains) at the same site, six varieties (strains) of *A. dahurica* var. *formosana* were also sown at one of the sites in Suining (Yongyi, Suining), and were named XD (BZA001), XE (BZA002), XF (BZA003), XG (BZA004), XH (BZB002) and XI (BZB003), respectively. All the plants were sown in late September of 2018. Superphosphate (750 kg/hm) and potassium phosphate (120 kg/hm) were applied as basal fertilizers. In December, the first topdressing of urea was applied at a rate of 66 kg/hm. The second topdressing of 165 kg/hm urea was carried out in February of the following year. The third topdressing was carried out in March of the following year, applying 99 kg/hm urea, 750 kg/hm superphosphate and 8 kg/hm potassium sulfate. Plots within the field were established in a randomized block design, with each plot consisting of four rows (5 m long, 3 m wide) and three replicated plots per treatment.

All *A. dahurica* var. *formosana* samples were collected at harvest (sown in September of the first year and harvested in July of the following year). A five-point sampling method was used, representative and robust *A. dahurica* var. *formosana* plants were selected, excess

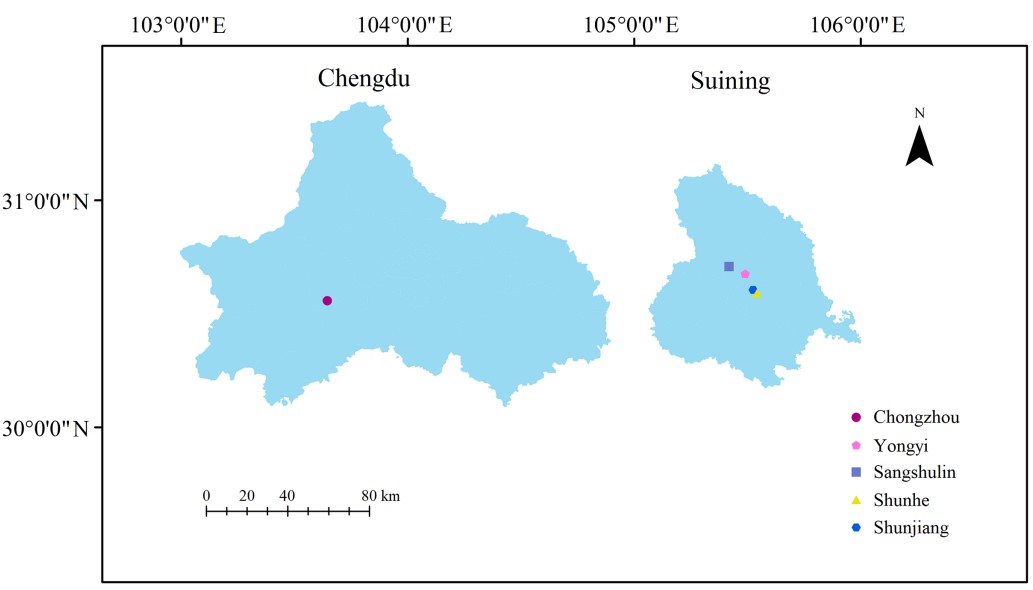

**Figure 1 Maps of the five different experimental sites.** Suining is the authentic production area for *A. dahurica* var. *formosana*, while Chongzhou is not.

bulk soil was shaken off, and the soil that remained attached to the roots was considered the rhizosphere soil (*Smalla et al., 1993*). Soil samples were preserved at −80 °C and 4 °C before use. Moreover, to avoid affecting yield estimates, the weight of sampled roots was recorded and included in the calculation of total yield.

## Soil physicochemical properties

Soil samples were collected from the rhizosphere of *A. dahurica* var. *formosana* from different experimental sites or varieties (strains) for the determination of physicochemical properties. The soil pH was measured with a pHB-8 pen-type acidity meter in a 1:2.5 soil/water (W/V) suspension (*Yang et al., 2018*). Organic matter (OM) was measured by the potassium dichromate volumetric method (*Ciavatta et al., 1991*). Soil total nitrogen (TN), total phosphorus (TP), and total potassium (TK) were determined according to Bahr's method (*Bahr, Chamba Zaragocin & Makeschin, 2014*). The hydrolysable nitrogen (HN) in the soil was determined by the alkali hydrolysis diffusion method (*Mulvaney & Khan, 2001*). Available phosphorus (AP) was obtained by using the sodium hydrogen carbonate solution-Mo-Sb anti-spectrophotometric method (*Hamer et al., 2013*). Available potassium (AK) was obtained by the ammonium acetate extraction-flame photometric method (*Shen et al., 2013*).

## Yield and active component levels of *A. dahurica* var. *formosana*

Yield was estimated by the ratio of the total fresh weight of the roots to the area in each plot. Representative roots of *A. dahurica* var. *formosana* with uniform growth were washed, dried at 105 °C for 15 min, and then dried at 55 °C to a constant weight. The contents of imperatorin and isoimperatorin of *A. dahurica* var. *formosana* were

determined according to the Chinese Pharmacopoeia method (2020 edition, Part I) (*Chinese Pharmacopoeia Commission, 2020*).

## Soil DNA extraction and MiSeq sequencing

The genomic DNA of soil samples was extracted by the CTAB or SDS method. The 16S rRNA of rhizosphere bacteria was amplified with 16S V3-V4 region primers (341F: CCTAYGGGRBGCASCAG, and 806R: GGACTACNNGGGTATCTAAT). PCR amplification was carried out using Phusion High-Fidelity PCR Master Mix with GC Buffer (New England Biolabs, Beijing, China) and high-fidelity enzymes to ensure amplification efficiency and accuracy. PCR products were detected by 2% (w/v) agarose gel electrophoresis, mixed in equal amounts according to concentration, and purified with a GeneJET Gel Extraction Kit (Thermo Fisher Scientific, Beijing, China). An Ion Plus Fragment Library Kit 48 rxns (Thermo Fisher Scientific, Beijing, China) was used for library construction. After Qubit quantification and library detection, Ion S5TMXL (Thermo Fisher Scientific, Beijing, China) was used for computer sequencing. Low-quality portions of the sequencing reads were cut using Cutadapt (V1.9.1, http://cutadapt.readthedocs.io/en/stable/) (*Aßhauer et al., 2015*). Sample data were separated from reads according to barcode, and barcode and primer sequences were truncated. The sequencing reads were compared by Vsearch and a species annotation database, and chimeric sequences were removed to obtain effective tags (*Rognes et al., 2016*). The raw reads were deposited in the NCBI Sequence Read Archive (SRA) database with accession number PRJNA742557.

## Statistical analysis

Sequences of clean reads with ≥97% similarity were assigned to the same OTUs (operational taxonomic units) using Uparse software (v8.1.1861) (*Edgar et al., 2011*). Species annotation analysis was carried out by Mothur and the SILVA database (Release 132), and the threshold value was set to 0.8~1 (*Edgar, 2004*; *Wang et al., 2007*; *Edgar, 2013*). The Observed species, Shannon, Chao1 and Goods coverage were calculated by QIIME (Version 1.9.1) (*Caporaso et al., 2010*). The WGCNA, stats, and ggplot2 packages of R (Version 3.4.1) were used to analyze beta diversity differences between groups and plot a PCoA chart. In the analysis of species differences between groups, LEfSe software (LEfSe 1.0) was used, and the LDA score was set to 4 by default. ADONIS was performed using the R vegan mrpp function and the adonis function. In Spearman's correlation analysis, the corr.test function of the psych package of R (Version 3.4.1; *R Core Team, 2017*) was first used to calculate Spearman's correlation values of species and environmental factors and test their significance, and then the heatmap function was used for visualization (*Algina & Keselman, 1999*).

## RESULTS

### The physicochemical properties of the rhizosphere soil of *A. dahurica* var. *formosana*

Overall, as shown in Fig. 2, the available nutrient levels of rhizosphere soil in Chongzhou were significantly higher than those in Suining at different experimental sites.
The rhizosphere soil pH at Chongzhou was neutral (7.1) and that of Suining was alkaline (pH values from 7.8–8.0) (Fig. 2A). Compared with previous results (Zhai et al., 2010), the pH of the rhizosphere soil was approximately 0.49–1.08 lower than that of the nonrhizosphere soil. The OM content of the Chongzhou rhizosphere soil was three- to 10 fold that of the other rhizosphere soils in Suining, up to 22.7 g/kg. The rhizosphere soil TN was the highest in Chongzhou (1.48 g/kg) and lowest in Yongyi in Suining (0.33 g/kg) (Fig. 2B). The HN, AP and AK of the rhizosphere soil of *A. dahurica* var. *formosana* from Chongzhou were significantly higher than those from Suining. Among the four experimental sites in Suining, soil HN and AP from Yongyi were the lowest (Fig. 2C).

Among all varieties (strains), there were significant differences in the soil nutrient levels of the *A. dahurica* var. *formosana* rhizosphere. There was little difference in pH among varieties (7.9–8.2) (Fig. 2D). The OM varied from 2.06–5.50 g/kg, among which BZA004 had the lowest amount (2.06 g/kg) (Fig. 2E). The TN, TP and TK values changed little. The rhizosphere soil AP of BZA002 was the highest (5.28 mg/kg), while that of BZA004 was the lowest (3.83 mg/kg). The AK value changed greatly from 31.73 to 123.94 mg/kg; BZA002 showed the highest value (123.94 mg/kg) and BZA004 showed the lowest (31.73 mg/kg) (Fig. 2F).

### Yield and active component levels of *A. dahurica* var. *formosana* from different experimental sites and varieties (strains)

For different experimental sites of *A. dahurica* var. *formosana*, the yield in Suining was approximately 3–4 times that in Chongzhou (Fig. 3A), although the active component levels of *A. dahurica* var. *formosana* from Suining were not as high as those from Chongzhou (Fig. 3B). Specifically, the yield of *A. dahurica* var. *formosana* from Yongyi was the highest (33,586.67 kg/hm) and that from Chongzhou was the lowest (9,166.67 kg/hm). The imperatorin content of *A. dahurica* var. *formosana* in Chongzhou was the highest (2.83 mg/g) and that in Yongyi was the lowest (1.50 mg/g). The isoimperatorin content in Shunjiang was the highest (1.34 mg/g), followed by that in Chongzhou (1.03 mg/g), and that in Yongyi was the lowest (0.85 mg/g). Yield × quality was used to comprehensively evaluate the samples, and it was found that the scores of the Suining experimental sites were significantly higher than those of Chongzhou (Fig. 3C). Therefore, *A. dahurica* var. *formosana* grown in an authentic production area is significantly better than that from other production areas.

For different varieties (strains) of *A. dahurica* var. *formosana* from the same experimental sites, the yield of BZA002 was the highest (43,043.33 kg/hm) and that of BZA004 was the lowest (30,463.33 kg/hm) (Fig. 3D). The imperatorin content of BZA004 was the highest (2.22 mg/g) and that of BZA001 was the lowest (1.50 mg/g).

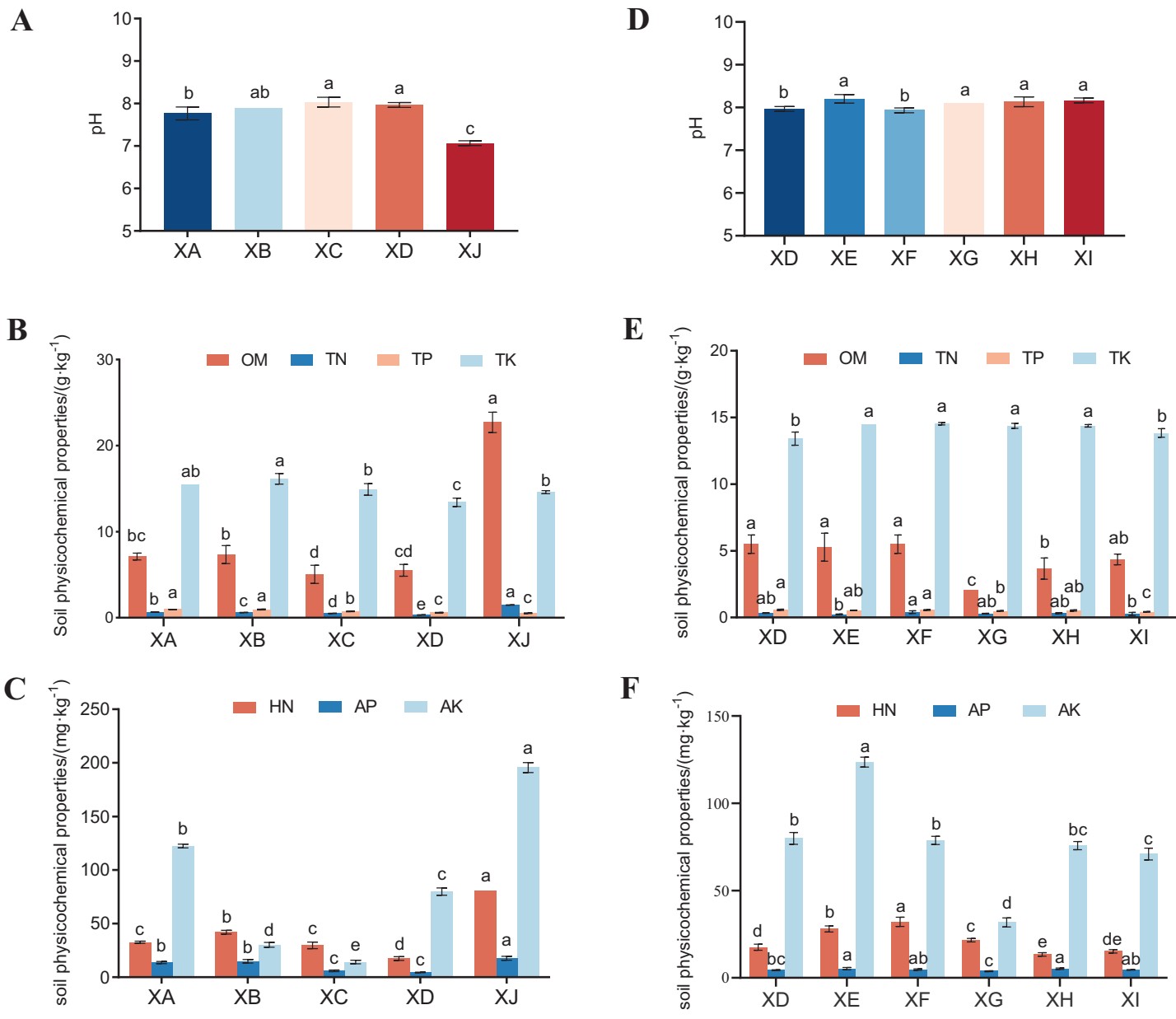

**Figure 2 The physicochemical properties of the rhizospheric soil.** (A) pH, (B) OM, TN, TP, TK, and (C) HN, AP, AK of the rhizosphere soil from different experimental sites; (D) pH, (E) OM, TN, TP, TK, and (F) HN, AP, AK of the rhizosphere soil from different varieties (strains). Different letters in the same color indicate significant differences ($P < 0.05$) among treatments based on Duncan's test. Abbreviation: OM, organic matter; TN, total nitrogen; TP, total phosphorus; TK, total potassium; HN, hydrolysable nitrogen; AP, available phosphorus; AK, available potassium.

The isoimperatorin content of BZA004 was the highest (1.08 mg/g) and that of BZB002 was the lowest (0.63 mg/g) (Fig. 3E). Yield × quality was used to comprehensively evaluate the samples, and it was found that the score of BZA002 was significantly higher than that of other varieties (strains) (Fig. 3F).

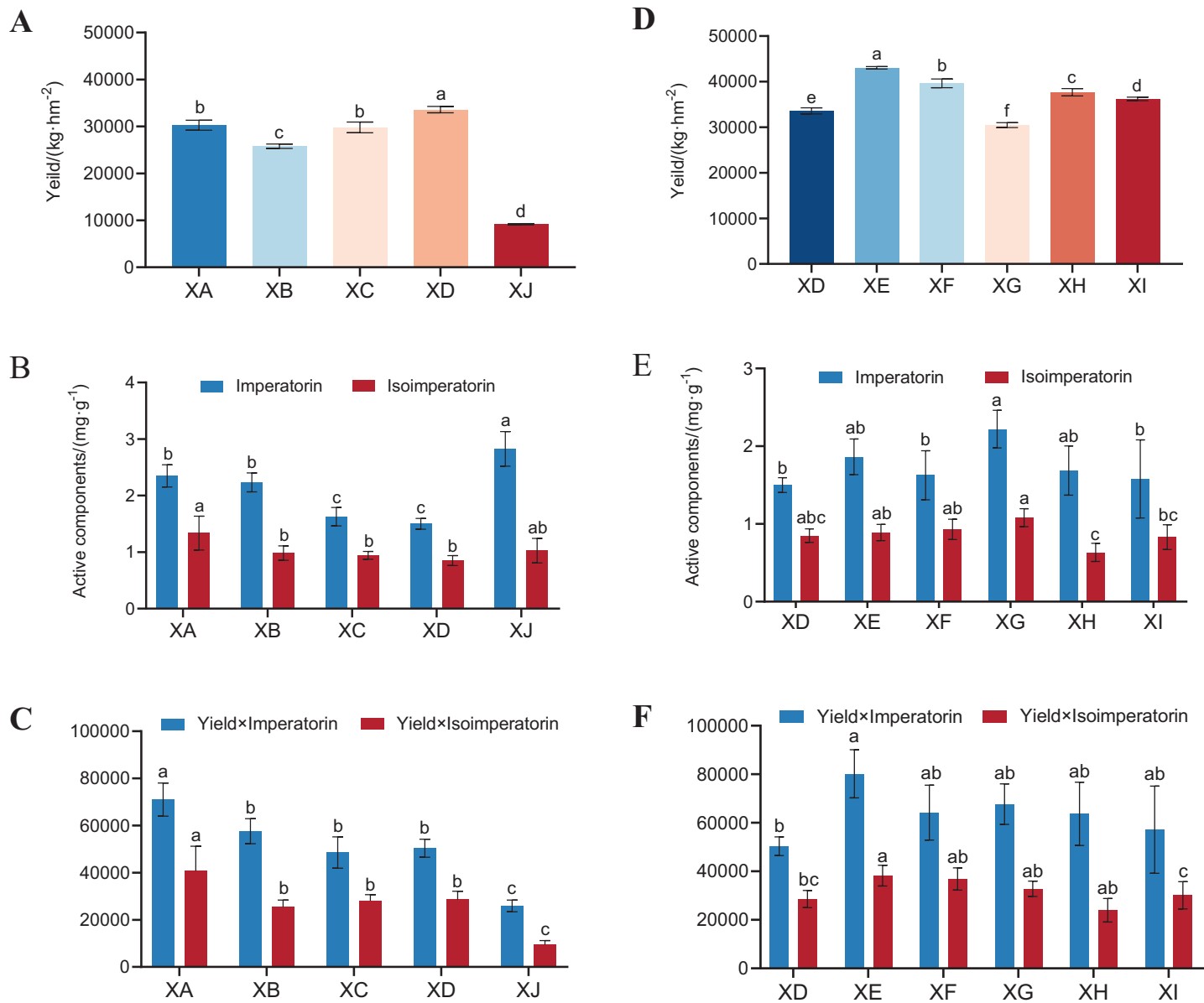

**Figure 3 The yields and active component levels of *A. dahurica* var. *formosana*.** The yields of *A. dahurica* var. *formosana* from (A) different experimental sites and (D) different varieties (strains); imperatorin and isoimperatorin in *A. dahurica* var. *formosana* from (B) different test sites and (E) different varieties (strains); the yield×imperatorin and yield×isoimperatorin of *A. dahurica* var. *formosana* from (C) different test sites and (F) different varieties (strains). Different letters in the same column indicate significant differences ($P < 0.05$) among treatments based on Duncan's mean test.

## Bacterial diversity in the rhizosphere soil of *A. dahurica* var. *formosana*

Through comparison with the SILVA database (Release 132), species annotation and statistical analysis of different classification levels, a total of 16,557 OTUs were found, of which 16,557 (100.00%) could be annotated to the database. As shown in Fig. 4, for the same varieties (strains), the Observed species, Chao1 and Shannon indices of rhizosphere bacteria from *A. dahurica* var. *formosana* in the Chongzhou experimental site with high nutrient levels and low yield were the lowest, while the other four experimental sites in

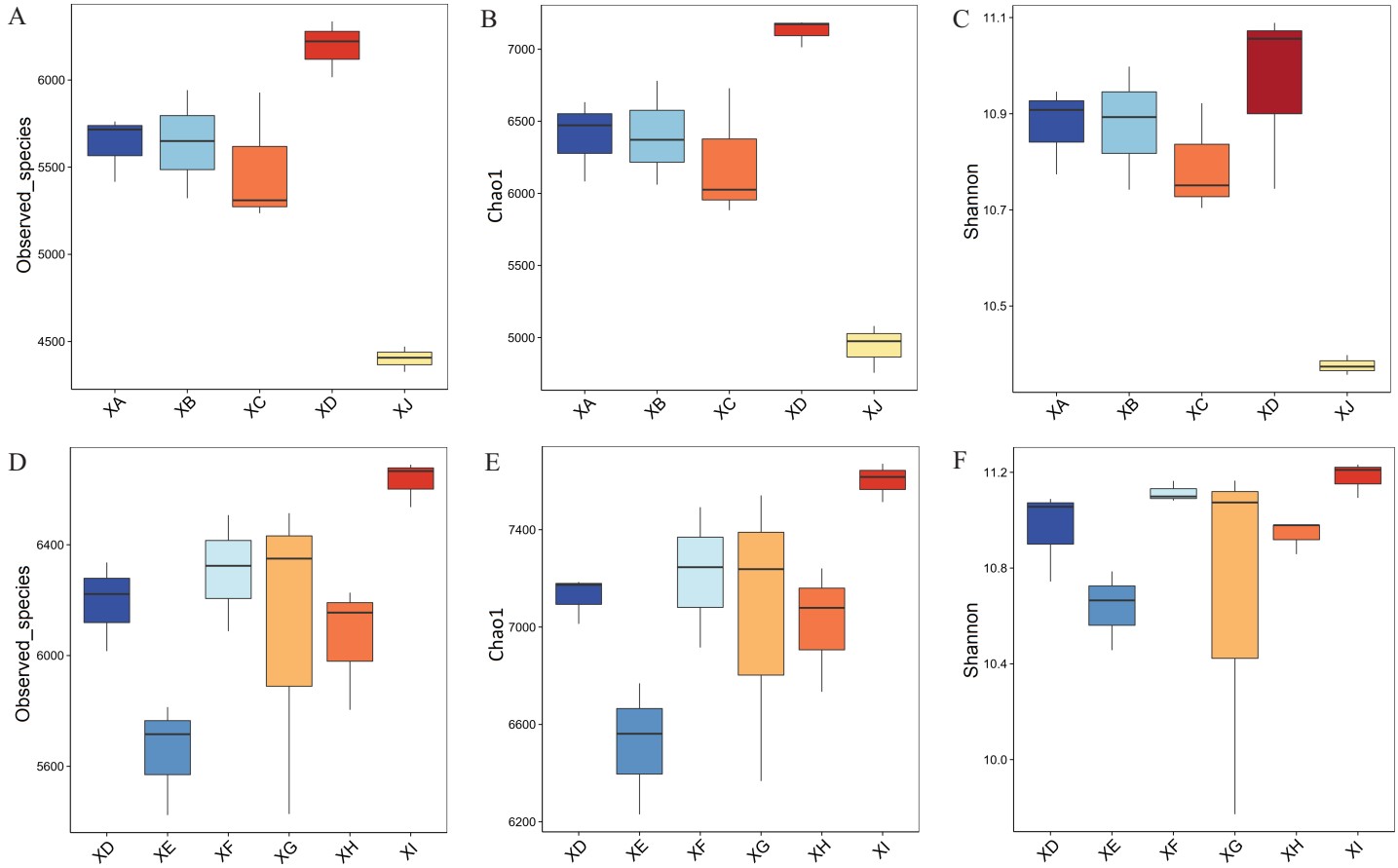

**Figure 4 Diversity indices of the bacterial community in rhizosphere soil samples.** (A) Observed species, (B) Chao1, and (C) Shannon indices of rhizosphere bacteria from different experimental sites. (D) Observed species, (E) Chao1, and (F) Shannon indices of rhizosphere bacteria from different varieties (strains).                                               

Suining had significantly higher index values. Among them, the Observed species and Chao1 indices of rhizosphere bacteria from Yongyi, which had the lowest nutrient levels, were the highest, indicating that rhizosphere bacteria from Yongyi had higher species richness and diversity. For different varieties (strains) of *A. dahurica* var. *formosana*, the Observed species and Chao1 indices of rhizosphere bacteria of BZA002 with high nutrient levels and yield were less than those of the others from Yongyi, while the bacterial diversity indices of BZB003 were the highest, indicating that the rhizosphere bacteria of BZB003 had higher species richness and diversity.

## Relative abundance of rhizosphere bacteria of *A. dahurica* var. *formosana*

In the rhizosphere soil of *A. dahurica* var. *formosana*, *Proteobacteria* was the dominant phylum. For different experimental sites, samples from Yongyi (XD) had the most abundant *Proteobacteria* sequences (49.1%) and the lowest abundances of *Acidobacteria* (6.6%) and *Rokubacteria* (1.1%). The proportions of *Firmicutes* and *Actinobacteria* in the rhizosphere soil samples of *A. dahurica* var. *formosana* from Chongzhou were significantly

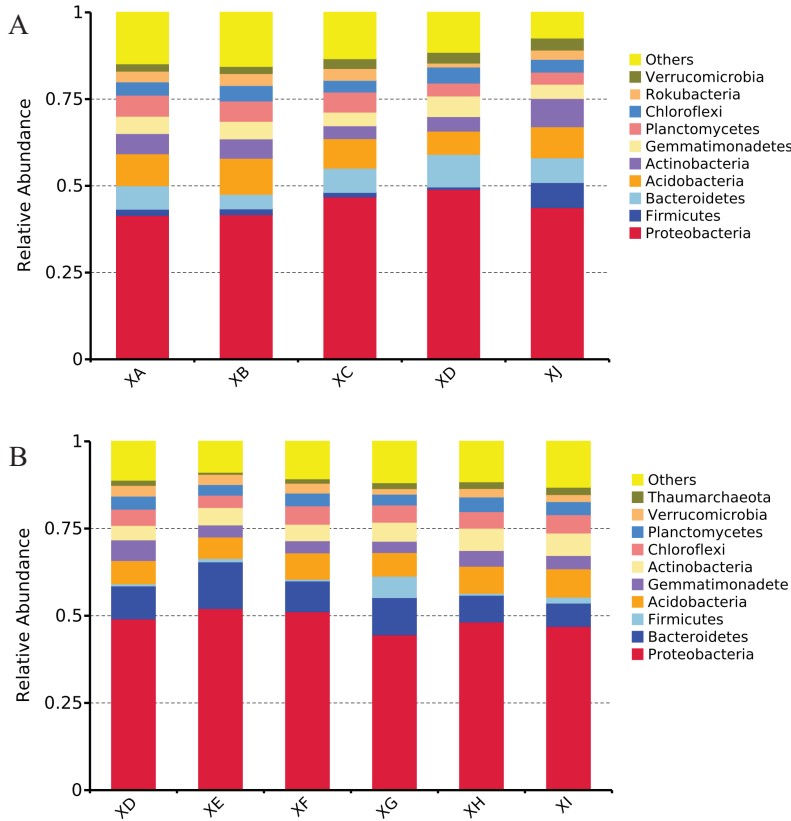

**Figure 5  Relative species abundance at the phylum level.** (A) Relative species abundance from different experimental sites. (B) Relative species abundance from different varieties (strains) (top 10 phyla).

higher than those at the other Suining sites, showing ratios of 7.2% and 8.1% (Fig. 5A). For different varieties (strains) of *A. dahurica* var. *formosana* from Suining, the proportion of *Proteobacteria* was highest in the rhizosphere of BZA002 (52.0%) and lowest in the rhizosphere of BZA004 (44.6%). The proportion of *Acidobacteria* was highest in the rhizosphere of BZB003 (8.1%) and lowest in the rhizosphere of BZA002 (6.1%). The proportion of *Actinobacteria* was highest in the rhizosphere of BZB002 and BZB003, showing ratios of 6.4%. The proportion of *Verrucomicrobia* was lowest in the rhizosphere of BZA004, only 1.6% (Fig 5B).

At the genus level, we selected 35 of the most abundant genera to generate a heatmap (Fig. 6). The results showed that the most abundant genera were *Bacteroides*, *Flavobacterium*, unidentified *Clostridiales*, *Anaeromyxobacter*, *Sphingomonas*, *Faecalibacterium*, *Polycyclovorans*, unidentified *Acidobacteria*, *Haliangium*, *Terrimonas*, *etc*. *Anaeromyxobacter*, *Sphingomonas*, *Polycyclovorans* and *Haliangium* belong to the phylum *Proteobacteria*.

For different experimental sites, a higher proportion of the genera *Haliangium* and *Anaeromyxobacter* and unidentified *Clostridiales* was detected in the rhizospheric bacteria of BZA001 from Chongzhou (XJ) with the highest nutrient levels (Fig. 6A). For different varieties (strains), *Sphingomonas*, *Flavobacterium* and *Terrimonas* accounted for a higher

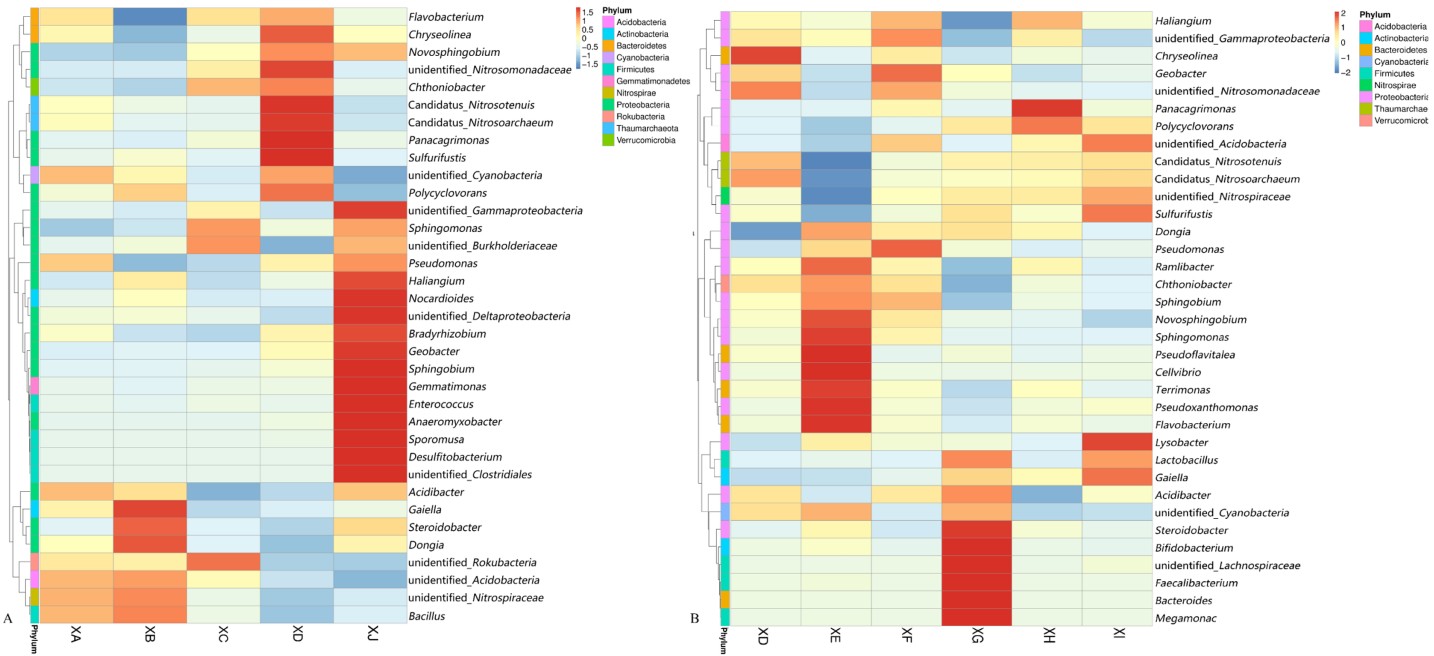

**Figure 6 Mapped images based on abundance information.** Cluster heatmap (A) from different sites and (B) different strains (top 35 genera). The more saturated the red color is, the higher the species abundance; the more saturated the blue color is, the lower the abundance of the genus. Abscissae indicate species annotation information, and ordinates indicate sample information. The clustering tree on the left side of the figure is a sample cluster tree.

proportion among the rhizosphere bacteria of high-yield BZA002 (XE). *Faecalibacterium* and *Bacteroides* accounted for a higher proportion among the rhizosphere bacteria of high-yield BZA004 (XG) (Fig. 6B).

## PCoA cluster analysis of the rhizosphere bacteria of *A. dahurica* var. *formosana*

Principal coordinate analysis (PCoA) was conducted among the treatment groups to explore the similarities and differences in the communities among the different groups, and the significance of the differences between the treatment groups was tested by nonparametric multivariate analysis of variance (ADONIS). The results showed some differences among the groups. For different experimental sites of *A. dahurica* var. *formosana*, the variances explained by principal component 1 and principal component 2 were 24.89% and 15.76%, respectively. The rhizosphere bacterial community of *A. dahurica* var. *formosana* in Chongzhou (XJ) was most different from that in other sites, followed by that in Yongyi in Suining (XD) (Fig. 7A). The results of ADONIS analysis once again indicated that there were significant differences between Suining and Chongzhou (Fig. 8). For different varieties (strains) of *A. dahurica* var. *formosana* from the same experimental site, the variances explained by principal component 1 and principal component 2 were 10.16% and 8.84%, respectively. There were some differences in rhizospheric bacterial communities among the different varieties (strains), and the rhizospheric bacterial communities of BZA002 and BZA004 were the most different from those of other varieties (strains) (Figs. 7B, 8). In this study, the results indicated that there
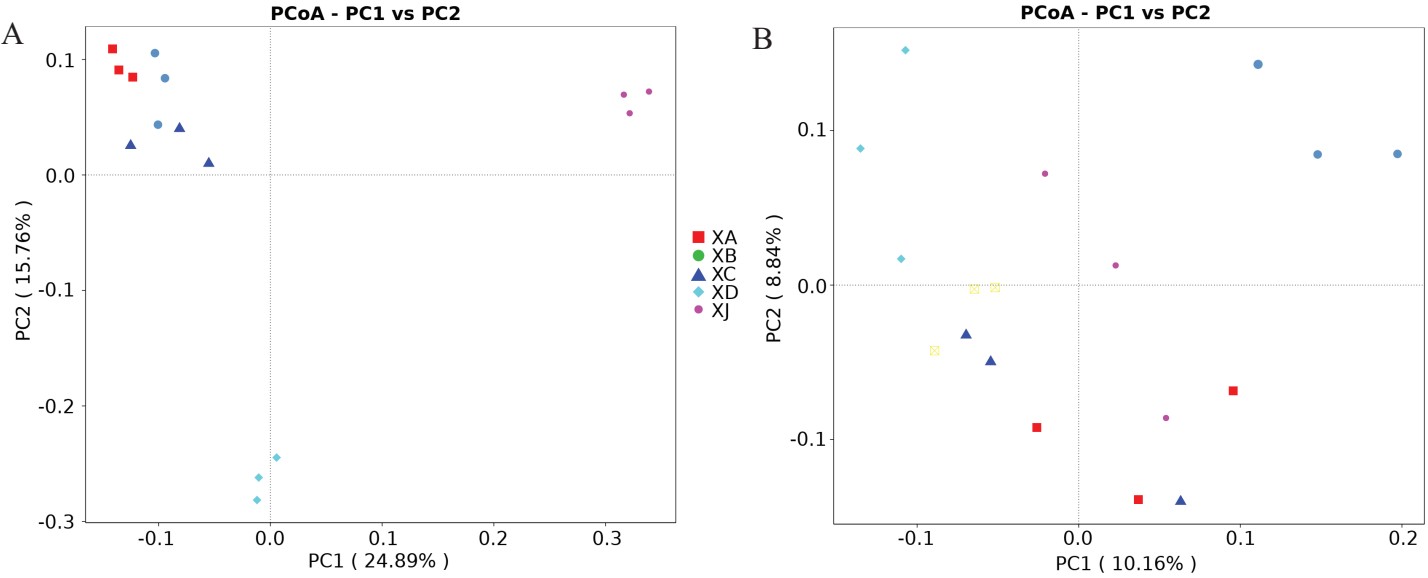

**Figure 7 Principal coordinate analysis (PCoA) of the rhizosphere bacterial community.** (A) PCoA for different planting areas and (B) PCoA for different genotypes.

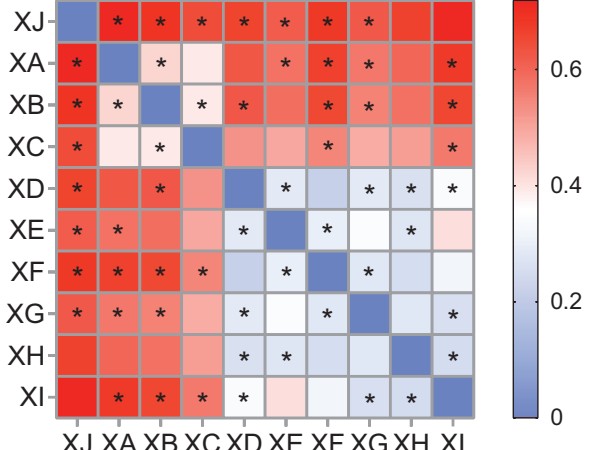

**Figure 8 Heatmap based on R² value of ADONIS of the rhizosphere bacterial community.** The darker the red, the larger the $R^2$ value, indicating that there are more grouping factors that can explain the sample differences. An asterisk (*) indicates a significant difference between groups ($P < 0.05$).

were significant differences among the experimental groups in the selected experimental sites and background varieties (strains). In addition, the experimental site had a greater effect on the rhizosphere bacterial community structure of *A. dahurica* var. *formosana* than the germplasm (Fig. 8).

## LEfSe analysis of the rhizosphere bacteria of *A. dahurica* var. *formosana*

Linear discriminant analysis effect size (LEfSe) analysis can identify biomarkers with significant differences between groups. The biomarkers with significant differences in the

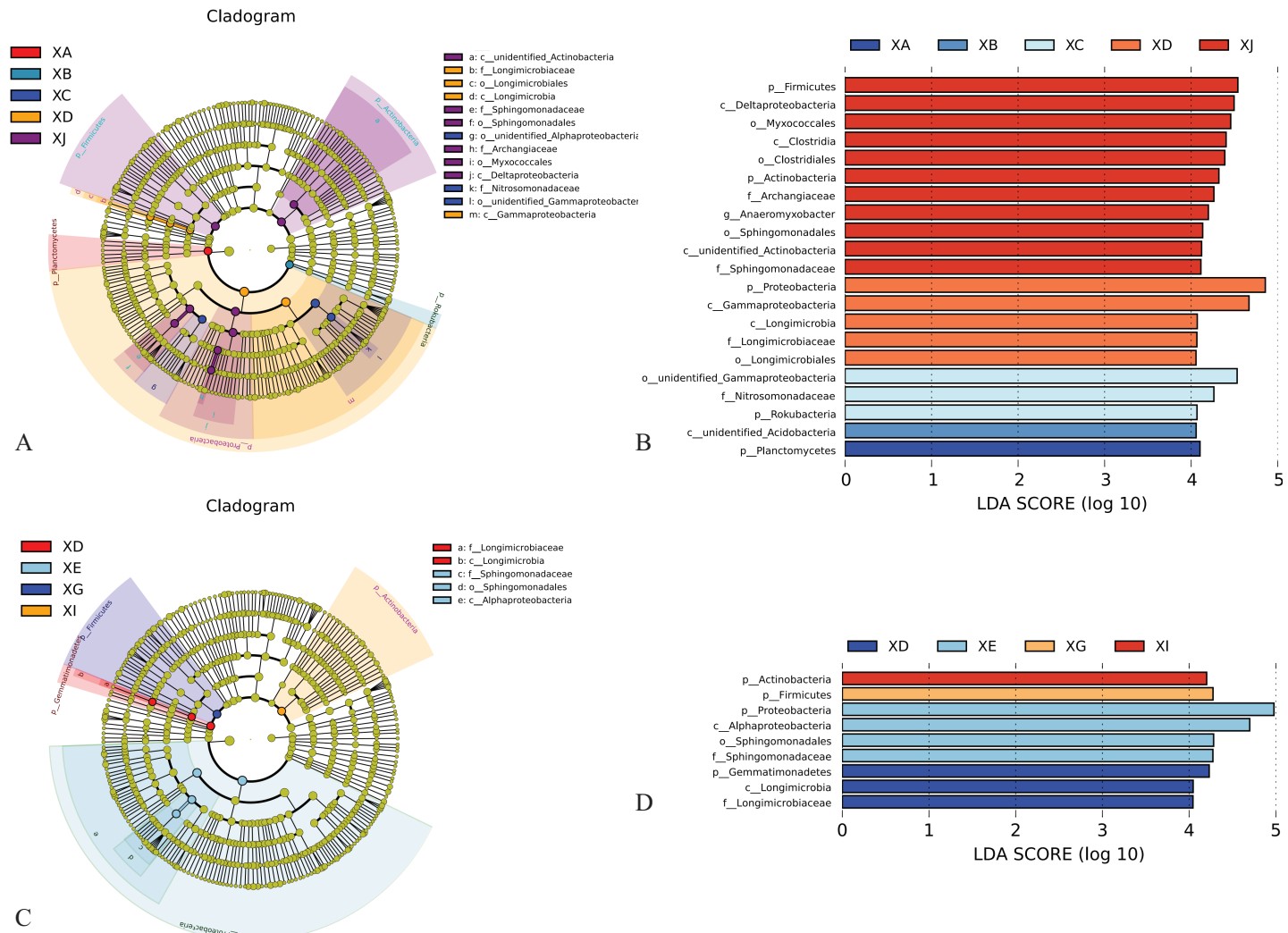

**Figure 9 Linear discriminant analysis effect size (LEfSe) method.** Significantly different abundant taxa from different experimental sites (A) and varieties (strains) (C) in the rhizosphere. The groups with significant differences in relative abundance among treatments are represented by colored dots. The size of the colored dots is proportional to the relative abundance. The Linear discriminant analysis (LDA) scores of the soil from different experimental sites (B) and varieties (strains) (D) in each treatment. The figure shows species with significant differences (LDA value > 4.0). Abbreviations: p, Phylum; c, Class; o, Order; f, Family; g, Genus; s, Species.

rhizosphere samples of *A. dahurica* var. *formosana* from different sites or different varieties (strains) were revealed through LEfSe analysis, and the linear discrimination analysis (LDA) values are given in Fig. 9. The results showed that there were 21 biomarkers with LDA scores >4. At the phylum level, for different experimental sites, enrichment of *Planctomycetes* was significant in the rhizosphere soil of BZA001 from Shunjiang (XA); enrichment of *Acidobacterira* was significant in the samples from Shunhe (XB); enrichment of *Roteobacteria* was significant in the samples from Sangshulin (XC); *Proteobacteria* contributed the most in the samples from Yongyi (XD), which had the highest yield of *A. dahurica* var. *formosana*, while *Firmicutes* and *Actinobacteria* contributed the most in the samples from Chongzhou (XJ), which had the highest active component levels in *A. dahurica* var. *formosana* (Figs. 9A, 9B). These biomarkers could

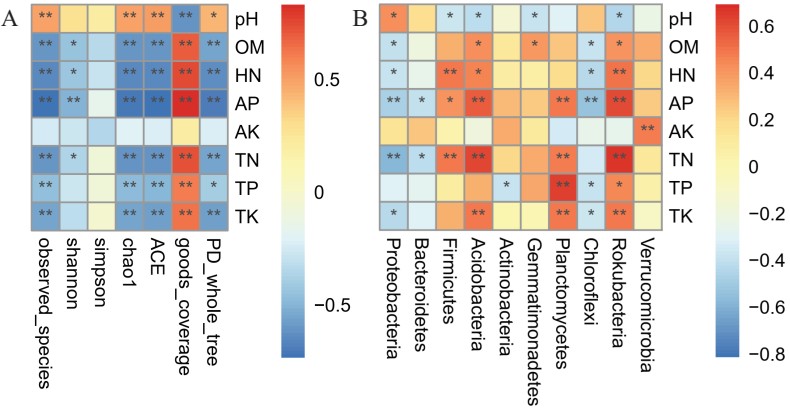

**Figure 10 Spearman's correlation heatmap analysis between soil factors and (A) diversity indices and (B) abundance of bacterial communities at the phylum level.** The vertical axis contains the environmental factor information, and the horizontal axis is the diversity index. The value corresponding to the middle heatmap is the Spearman's correlation coefficient r, which is between −1 and 1. An asterisk (*) indicates a significant correlation (P < 0.05). Two asterisks (**) indicate an extremely significant correlation (P < 0.01). Abbreviations: OM, organic matter; TN, total nitrogen; TP, total phosphorus; TK, total potassium; HN, hydrolysable nitrogen; AP, available phosphorus; AK, available potassium.

represent the key bacterial taxa shaping the rhizosphere bacterial community of *A. dahurica* var. *formosana*.

From the same experimental sites, the major divergent species of different varieties (strains) of the *A. dahurica* var. *formosana* rhizosphere soil also differed. There were nine biomarkers with LDA scores >4. For example, *Gemmatimonadetes* contributed the most to the bacterial community of the BZA001 rhizosphere soil (XD); *Proteobacteria* contributed the most to the bacterial community of the BZA002 rhizosphere soil (XE), which had the highest yield of *A. dahurica* var. *formosana*; *Firmicutes* contributed the most to the bacterial community of BZA004 rhizosphere soil (XG), which had the highest active component level of *A. dahurica* var. *formosana*; *Actinobacteria* contributed the most to the bacterial community of BZB003 rhizosphere soil (XI) (Figs. 9C, 9D).

Combined with the results of the rhizospheric bacteria from the five experimental sites or six varieties (strains), it was found that *Proteobacteria* had a significant influence on the rhizosphere bacterial community of the samples with high yields, and *Firmicutes* had a significant influence on the rhizosphere bacterial community of the samples with high contents of active components, regardless of whether the influencing factors were from differences in experimental sites or varieties (strains). In addition to the phylum level, at the family level, *Sphingomonadaceae* was found to be significantly enriched in soils with high nutrients and low alpha diversity.

## Correlation analysis between environmental physicochemical factors and bacterial taxa

Spearman's correlation analysis with soil nutrients showed that, except for AK, the rhizosphere diversity of *A. dahurica* var. *formosana* was significantly negatively correlated with soil nutrients (Fig. 10A). Moreover, the physicochemical properties of the rhizosphere

soil, especially AP and TN, which exerted the greatest influence on microbial community composition, were closely related to bacterial taxa (Fig. 10B). The abundances of *Acidobacteria* (r = 0.5694) and *Rokubacteria* (r = 0.6158) were most closely related to AP, and those of *Acidobacteria* (r = 0.6209) and *Rokubacteria* (r = 0.6653) were also most closely related to TN. Moreover, the abundance of *Proteobacteria* was significantly negatively correlated with AP (r = −0.4811) and TN (r = −0.5790), and the abundance of *Firmicutes* was significantly positively correlated with HN (r = 0.4895) and TN (r = 0.4927). The abundance of *Actinobacteria* was only negatively correlated with TP (r = −0.3783), while the abundance of *Verrucomicrobia* was positively correlated with AK (r = 0.4758).

## DISCUSSION

In this study, the rhizosphere bacterial diversity of *A. dahurica* var. *formosana* with high active components was low, and imperatorin may be one of the reasons for the decrease in rhizosphere microbial diversity. The main active components of *A. dahurica* var. *formosana* are coumarins and essential oils, and the coumarins mainly include imperatorin and isoimperatorin (*Shi et al., 2022*). Regardless of the difference in production area or germplasm, the variation in imperatorin occurred in a direction opposite to that of bacterial diversity. To our knowledge, coumarins in *A. dahurica* var. *formosana* have antibacterial activity, with inhibitory effects on a variety of pathogens (*Bhattarai et al., 2021*; *Yang et al., 2022*).

*Proteobacteria* members were the dominant rhizosphere bacteria of *A. dahurica* var. *formosana*. These eutrophic bacteria can mineralize soil nutrients, enhance the absorption of nutrients by plants, and promote plant growth (*Tian et al., 2020*). This is similar to observations from the dominant rhizosphere communities of other medicinal plants, such as healthy *Panax notoginseng* and *Bupleurum chinese* (*Tan et al., 2017*; *Liu et al., 2022*). Regardless of the difference in experimental site or variety (strain), *Proteobacteria* was the dominant phylum in the rhizosphere of *A. dahurica* var. *formosana*, and the dominant genera were also similar. The results indicated that the rhizosphere bacterial community of *A. dahurica* var. *formosana* is stable and conserved and could adapt to different environmental changes and resist interferences from external factors to a certain extent, providing a theoretical basis for planting in different locations and expanding the zones of production.

*Acidobacteria* and *Rokubacteria* were prevalent in the rhizosphere of *A. dahurica* var. *formosana*. There are a variety of clusters for the genes associated with the synthesis of antibiotics, antifungins, ferritin and immunosuppressants in these bacteria (*Crits-Christoph et al., 2018*). In the similar environment of the four experimental sites in Suining, the bacterial diversity of XD was high when the abundance of *Acidobacteria* and *Rokubacteria* in the rhizosphere was low, suggesting that these bacteria with antibacterial activity might inhibit the colonization of other bacteria in the rhizosphere of *A. dahurica* var. *formosana*. Due to continuous cropping obstacle, the rhizosphere microflora of *P. notoginseng* and other peanut species change from a "bacterial-type" to "fungal-type" during continuous cropping periods, and the bacterial community of *Proteobacteria* and other bacteria show a downward trend (*Tan et al., 2017*; *Li et al., 2014*). There were no

obvious continuous cropping obstacles in *A. dahurica* var. *formosana*, which might be due to the inhibition of soil-borne diseases by its antibacterial active components and bacterial community.

A high abundance of *Proteobacteria* was an important rhizospheric indicator of high yield, and a high abundance of *Firmicutes* was an important indicator of high quality. Yield and quality are the result of many factors in plant growth (*Das et al., 2022*). Regardless of the difference in experimental site or variety (strain), based on LEfSe analysis, *Proteobacteria* were consistently significantly enriched in the rhizosphere of the high-yield samples, while *Firmicutes* were consistently significantly enriched in the rhizosphere of the samples with high active component levels. The results indicate that abundant *Proteobacteria* bacteria may be a feature of the rhizosphere bacterial community conducive to the growth and development of *A. dahurica* var. *formosana*, and abundant *Firmicutes* may be a rhizosphere feature conducive to the accumulation of effective components. Niraula suggested that differences in soybean yield might be attributed to variations in rhizosphere microbes at taxonomic, functional, and community levels (*Niraula, Rose & Chang, 2022*). Many rhizosphere growth-promoting bacteria from *Proteobacteria* and *Firmicutes* can promote plant growth and mineral nutrient absorption (*Zhang, Vivanco & Shen, 2017*; *Tian et al., 2020*). Although the exact interaction between the beneficial bacterial community and *A. dahurica* var. *formosana* is unclear, high yield and high quality may be related to the cumulative rhizosphere bacterial community and cannot be attributed simply to a single bacterium or function. Therefore, we believe that promoting the abundance of *Proteobacteria* and *Firmicutes* and maintaining a beneficial rhizosphere bacterial community for *A. dahurica* var. *formosana* may play a potential role in promoting yield and quality.

Interestingly, *Sphingomonadaceae* was found among the differential taxa with LDA scores >4 for soil environmental differences or genotypic differences. *Sphingomonadaceae* was significantly enriched in the samples with the highest soil nutrient levels and lowest bacterial diversity. Many strains of *Sphingomonadaceae* can degrade polycyclic or monocyclic aromatic compounds, such as benzoic acid and salicylic acid, and use these aromatic compounds as carbon sources for growth (*Gou et al., 2015*). Allelopathic rice could secrete more aromatic compounds to induce the enrichment of these bacteria under chemical chemotaxis (*Lin, 2010*; *Feng et al., 2021*). *A. dahurica* var. *formosana* contains a large number of aromatic compounds (*Zhao et al., 2022*). A similar allelopathy may exist in *A. dahurica* var. *formosana*. An increase in active components in *A. dahurica* var. *formosana* caused more aromatic compounds to be secreted from the roots, recruiting more *Sphingomonadaceae* and having antibacterial effects. Further studies on root exudates of *A. dahurica* var. *formosana* should be carried out to verify this hypothesis.

Origin and germplasm had a significant influence on the structure of the rhizosphere bacterial community of *A. dahurica* var. *formosana*. The assembly of the rhizosphere microbial community is controlled by complex interactions between microorganisms, plant hosts, and the environment (*Chaparro, Badri & Vivanco, 2014*; *Trivedi et al., 2020*). PCoA indicated that the rhizosphere bacterial community of *A. dahurica* var. *formosana* showed significant differences among the different experimental sites and varieties

(strains). This confirmed the importance of suitable origins and germplasm for the cultivation of medicinal plants from the perspective of rhizosphere bacteria, which is of great significance for improving the yield and quality of *A. dahurica* var. *formosana*.

Compared with germplasm resources, a suitable planting site was a key factor in determining the assembly of the rhizosphere microbial community of *A. dahurica* var. *formosan*, which is of great significance for future approaches to improve yield and quality. The four sites in Suining were all located along the coast of the Fujiang River and shared a similar climate. Except for the Chongzhou test site (XJ), which had a different climate, the differences among experimental sites were slightly greater than the differences among varieties (strains). These results indicated that the rhizosphere bacterial community of *A. dahurica* var. *formosana* was particularly affected by soil parameters in this study. Previous studies have shown that for many plants, genotype has a significant influence on the bacterial community in the aboveground parts, especially in perennial plants, while the rhizospheric bacterial community is mostly affected by soil factors (*Wagner et al., 2016*). In sorghum, variety had little effect on the rhizosphere bacteria, while soil was the main factor (*Schlemper et al., 2017*). Automated ribosomal intergenic spacer analysis (ARISA) was used to investigate nine rice varieties treated with different levels of nitrogen. It was found that variety had a significant effect on the bacterial communities of the stems and buds of rice, while the bacterial community of the roots was mainly affected by the nitrogen levels in the paddy fields (*Sasaki et al., 2013*). Similarly, the rhizosphere bacterial community of *A. dahurica* var. *formosana* was significantly affected by the cultivation soil.

Except for AK, soil physicochemical properties were negatively correlated with bacterial diversity. Mutualism occurs between plants and a variety of microorganisms and provides more than half of the plant's nutritional needs (*van der Heijden et al., 2016*). Chaudhary found that rhizospheric bacteria could significantly improve the utilization rate of soil N, P and K, alleviating the deficiency of soil nutrients to a certain extent (*Chaudhary, Rathore & Sharma, 2020*). To our knowledge, when the phosphorus content in a soil is very low, legumes stimulate the enrichment of AMF in the rhizosphere by secreting isoflavones to promote nutrient absorption (*Hassan & Mathesius, 2012*). Moreover, when soil phosphorus content is very high, plants can obtain optimal nutrient supplies without the aid of AMF symbiosis (*Ferrol, Azcón-Aguilar & Pérez-Tienda, 2019*). Therefore, ensuring bacterial diversity in the rhizosphere of *A. dahurica* var. *formosana* may be a new approach for reducing fertilizer use.

In addition to differences in soil nutrients, the original soil bacterial community might be another soil factor that affects the rhizosphere bacteria of *A. dahurica* var. *formosana*. When *de Ridder-Duine et al. (2005)* studied the composition of rhizosphere and nonrhizosphere bacteria in different soils, he found that the structure of the rhizosphere bacterial community was largely determined by the nonrhizosphere soil bacterial community. In other words, the difference of original soil bacterial community in each experimental site might play an important role in the assembly of the characteristic rhizosphere bacterial community of *A. dahurica* var. *formosana* in each site.

## CONCLUSIONS

In general, these research results might help to improve the cultivation and management of *A. dahurica* var. *formosana*. The rhizosphere bacteria of *A. dahurica* var. *formosana* were stable and conserved and could adapt to different environmental and genetic changes to a certain extent, providing a theoretical basis for planting in different locations and expanding planting zones. High bacterial abundance of *Proteobacteria* and *Firmicutes* is an important characteristic of the rhizosphere of *A. dahurica* var. *formosana* for achieving high yield and quality, respectively. Increasing the abundance of *Proteobacteria* and *Firmicutes* and maintaining a favorable rhizosphere bacterial community structure may be a new approach for increasing production and quality and reducing the application of pesticides and fertilizers. In addition to *Proteobacteria* and *Firmicutes*, there are many bacterial taxa potentially related to the growth and accumulation of secondary metabolites in *A. dahurica* var. *formosana*, and these bacterial resources need to be further explored. Additionally, for the first time from the perspective of rhizosphere bacteria, the importance of suitable production areas and germplasm resources for the cultivation of *A. dahurica* var. *formosana* was verified, and the key role of the production area in assembling a beneficial microbiome was establish, which lays a theoretical foundation for expanding planting areas in the future.

### Funding

This work was supported by the Sichuan Science and Technology Program (No. 2021YFYZ0012), the Joint Implementation of Key R & D Projects in Sichuan and Chongqing in 2020 (No. 2020YFQ0054), the Foundation on Double-Support Plan of Disciplinary Construction in Sichuan Agricultural University-Innovation Team Projects (Grant No. P202108), and the Foundation on Sichuan Key Discipline Construction Project of Traditional Chinese Medicine (Grant No. 2021-16-4). The funders had no role in study design, data collection and analysis, decision to publish, or preparation of the manuscript.

### Grant Disclosures

The following grant information was disclosed by the authors:
Sichuan Science and Technology Program: 2021YFYZ0012.
Joint Implementation of Key R & D Projects in Sichuan and Chongqing: 2020YFQ0054.
Foundation on Double-Support Plan of Disciplinary Construction in Sichuan Agricultural University-Innovation Team Projects: P202108.
Foundation on Sichuan Key Discipline Construction Project of Traditional Chinese Medicine: 2021-16-4.

### Competing Interests

The authors declare that they have no competing interests.

## Author Contributions

- Meiyan Jiang conceived and designed the experiments, performed the experiments, prepared figures and/or tables, authored or reviewed drafts of the article, and approved the final draft.
- Fei Yao performed the experiments, prepared figures and/or tables, and approved the final draft.
- Yunshu Yang performed the experiments, prepared figures and/or tables, and approved the final draft.
- Yang Zhou analyzed the data, authored or reviewed drafts of the article, and approved the final draft.
- Kai Hou analyzed the data, authored or reviewed drafts of the article, and approved the final draft.
- Yinyin Chen performed the experiments, prepared figures and/or tables, authored or reviewed drafts of the article, and approved the final draft.
- Dongju Feng analyzed the data, prepared figures and/or tables, authored or reviewed drafts of the article, and approved the final draft.
- Wei Wu conceived and designed the experiments, authored or reviewed drafts of the article, and approved the final draft.

## Data Availability

Data are available in the NCBI Sequence Read Archive (SRA) database: PRJNA742557 (SRR15020834 to SRR15020863).

## Supplemental Information

Supplemental information for this article can be found online at http://dx.doi.org/10.7717/peerj.15997#supplemental-information.

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
