# Peer review of "Analysis of the rhizosphere bacterial diversity of Angelica dahurica var. formosana from different experimental sites and varieties (strains)"

_PeerJ, doi:10.7717/peerj.15997_

## Round 0.1 · original submission · Major Revisions

Dear Authors,

Many thanks for choosing PeerJ for your work. Two reviewers have gone through your manuscript and provided professional feedbacks about your study. Please revise your manuscript based on reviewers' comments and re-submit your work with point-by-point responses. We look forward to the submission of your revised work. Please be notified that your manuscript might be sent out for review again.

·

Basic reporting

At first I would to mention the actual topic of the study – to describe the microbial community of samples of Angelica dahurica from different sites or different varieties. The main point of the study is to determine the impact of site or variety (strain) to the plant yield and describe the related microbial taxa.
It seems that study was carried out several years ago with Biorxiv preprint available. There are strong reasons to use amplicon sequence variants (ASVs) instead of OTUs for less noise and avoid spurious taxons (https://www.mdpi.com/2306-5354/9/4/146).
Callahan, McMurdie and Holmes points that “the improvements in reusability, reproducibility and comprehensiveness are sufficiently great that ASVs should replace OTUs as the standard unit of marker-gene analysis and reporting” Callahan, B., McMurdie, P. & Holmes, S. Exact sequence variants should replace operational taxonomic units in marker-gene data analysis. ISME J 11, 2639–2643 (2017). https://doi.org/10.1038/ismej.2017.119

Experimental design

Some details of the study must be clarified to better understanding and reproducibility:
• L160: the FASTQ data for each sample presented as 3 separate gzipped files. Which of them paired and single?
• L163: BioProject PRJNA742557 is not available in public. Despite the reviewer access to the NCBI DataView I cannot dump the FASTQ data.
• L165/L161: the filtering procedure lacking details, only the reference is given. The filtering settings and software parameters are required.
• L167: SSUrRNA database version is required.
• Table 3: XD sample is duplicated

Validity of the findings

In introduction authors stands three objectives, the last of them is “(iii) to determine the contribution of experimental sites and varieties (strains) to the construction of a beneficial bacterial community and confirm the importance of good production areas and good germplasm resources in cultivation from the point of rhizosphere bacteria.”, but Discussion section says about different taxa contribution only (related to sites/varieties).
For better Results (L202) understanding I recommend to present Tables 1, 2, 3 as diagrams in relative scale. The additional column in Table 2 (yield × quality) can help to evaluate the overall rank of each sample. Table 4 looks better as the heatmap.
Why the XD sample has the different composition at Figure 1, 3?
L191: “BZA002 had the highest rhizosphere nutrients, while BZA004 had the lowest.” – according to the Table 1, the highest value of OM has BZA001 (in different sites). I think it’s hard to tell about variety in common, regardless to the site of growing. Please clarify the statement.
L288: What the meaning of the LDA score of 4? Are there any reasons for such threshold?
My notes about the main text:
• L219, Table 3: “shannon” → “Shannon”; “chao1” → “Chao1”
• L217: the SILVA database is not mentioned in Methods section.
• L448-450: the last sentence might be omitted.

Reviewer 2 ·

Basic reporting

The paper use clear, unambiguous, technically correct text. Most part of the article conform to professional standards of courtesy and expression.The language and grammar of the article including the title requires revisions.

Experimental design

Line128-129,There are no necessary to return the roots to the soil, just need to calculate the total roots weight.please express exactly.
Line144-145, Need the roots to be washed at 105°C for 15 minutes? Here should add one comma after ‘washed’. please express exactly.
Line 191-192 Which properties the rhizosphere nutrients include?Why BZA002 had the highest rhizosphere nutrients? Please explain.

Validity of the findings

The data on which the conclusions are based made available in an acceptable discipline-specific repository. The data was robust, statistically sound, and controlled.
However, the test site is one of the main factors. In order to tell readers the distribution area, I suggest to give a map of the different test sites.

---

## Round 0.2 · Minor Revisions

Dear authors,

The reviewers require certain minor revisions for your manuscript before it could be considered for publication by PeerJ. Please follow the reviewers' comment and revise it with point-by-point responses. Thanks for your efforts.

Kind regards,
Prof. Liang Wang, PhD

·

Basic reporting

pass

Experimental design

pass

Validity of the findings

correct

Additional comments

Notes about the main text:

L25: "is" to "are"
L127: which year?
L168: "Barcode" to "barcode" (?)
L182: there are number of programs to perform LEfSe analysis, reference required
L185: R version 2.15.3 is 10 years old, is it correct version specified?
L409: "A. dahurica var. formosana contains a large number of aromatic compounds" I suggest to add a reference
L453: "De Ridder-Duine" to "de Ridder-Duine"
L456-458: that is almost obvious sentence, that soil community plays a key role for rhizosphere microbial community composition
L461: I suggest active voice here "will be helpful in improving" to "might help to improve"

Manuscript requires proofreading after authors' extensive editing.

Reviewer 2 ·

Basic reporting

The paper use clear, unambiguous, technically correct text. Most part of he article conform to professional standards of courtesy and expression.The paper include sufficient introduction and background to demonstrate how the work fits into the broader field of knowledge.The structure of the article basically conform to an acceptable format of ‘standard sections’.Figures in this paper is relevant to the content of the paper, of sufficient resolution, and appropriately described and labeled.

Experimental design

The submission clearly define the research question, which was relevant and meaningful. The knowledge gap being investigated was identified clearly.

Methods described in this article had sufficient information to be reproducible by another investigator.

Validity of the findings

The data on which the conclusions are based made available in an acceptable discipline-specific repository. The data was robust, statistically sound, and controlled.

Additional comments

The paper has been greatly improved and is worthy of publication.

---

## Round 0.3 · Minor Revisions

Dear Authors,

Could you please revise the manuscript by following the reviewer's comments? It is a requirement that the response should be point-by-point.
* * *
Reviewer's Comments
The rebuttal letters for rev. 2 and rev. 3 (current) are identical. I cannot find any changes in the new version of the manuscript. My comments stay the same.

L25: "is" to "are"
L127: which year?
L168: "Barcode" to "barcode" (?)
L182: there are number of programs to perform LEfSe analysis, reference required
L185: R version 2.15.3 is 10 years old, is it correct version specified?
L409: "A. dahurica var. formosana contains a large number of aromatic compounds" I suggest to add a reference
L453: "De Ridder-Duine" to "de Ridder-Duine"
L456-458: that is almost obvious sentence, that soil community plays a key role for rhizosphere microbial community composition
L461: I suggest active voice here "will be helpful in improving" to "might help to improve"

Manuscript requires proofreading after authors' extensive editing.
* * *
A fluent English speaker should go through the manuscript and make sure that all language issues are solved. Otherwise, please use language editing services provided by a professional team.

·

Basic reporting

pass

Experimental design

pass

Validity of the findings

valid

Additional comments

The rebuttal letters for rev. 2 and rev. 3 (current) are identical. I cannot find any changes in the new version of the manuscript. My comments stay the same.

L25: "is" to "are"
L127: which year?
L168: "Barcode" to "barcode" (?)
L182: there are number of programs to perform LEfSe analysis, reference required
L185: R version 2.15.3 is 10 years old, is it correct version specified?
L409: "A. dahurica var. formosana contains a large number of aromatic compounds" I suggest to add a reference
L453: "De Ridder-Duine" to "de Ridder-Duine"
L456-458: that is almost obvious sentence, that soil community plays a key role for rhizosphere microbial community composition
L461: I suggest active voice here "will be helpful in improving" to "might help to improve"

Manuscript requires proofreading after authors' extensive editing.

---

## Round 0.4 · accepted · Accept

The authors have addressed the concerns pointed out by the reviewer with a point-by-point response. Therefore, I would like to recommend the acceptance of the manuscript.